# Primary care need and engagement by people with criminal legal involvement: Descriptive and associational analysis using retrospective data on the entire population ever detained in one southeastern U.S. county jail 2014–2020

**Michele M. Easter** [1]*, **Nicole L. Schramm-Sapyta** [2], **Marvin S. Swartz** [1], **Maria A. Tackett** [3], **Lawrence H. Greenblatt** [4]

1 Department of Psychiatry & Behavioral Sciences, Duke University School of Medicine, and Wilson Center for Science & Justice, Duke University School of Law, Durham, NC, United States of America, 2 Duke Institute for Brain Sciences, Duke University, Durham, NC, United States of America, 3 Department of Statistical Science, Duke University, Durham, NC, United States of America, 4 Department of Medicine, Duke University School of Medicine, Durham, NC, United States of America

* michele.easter@duke.edu

## Abstract

More than 7 million people are released each year from U.S. jails or prisons, many with chronic diseases that would benefit from primary care in their returning communities. The objective of this study was to provide an in-depth, payer-agnostic description and associational analysis of primary care need and utilization by all individuals ever detained in one county detention facility over a 7-year period. Detention records 2014–2020 were merged with data from an electronic health record with excellent coverage of local primary care, emergency, and hospital services. We found low primary care participation for the group as a whole, with under three quarters of those with serious chronic diseases ever seeing a primary care provider over a 7-year period and less than half ever having a year with more than one visit. Multivariable regression models estimated associations between individual characteristics (demographic, detention-related, and clinical) and ever having access to primary care (logistic) and the number of primary care visits (zero-inflated negative binomial). We found that having more jail bookings was associated with fewer primary care visits, but not one-time access, even controlling for time out of community, age, insurance, and other demographic characteristics. This finding was driven by subgroups with chronic disease such as hypertension, obstructive lung disease, and diabetes, who most need regular primary care. Being Black retained an independent effect, even controlling for bookings and other variables, and was also associated with fewer primary care visits, though not one-time access. To promote primary care utilization among individuals who have the combined challenges of repeated jail involvement and chronic disease, it is crucial to focus on engagement, as much as formal access. Access to health insurance alone will not resolve the problem; Medicaid expansion should be coupled with specialized, tailored support to promote engagement in primary care.

**Data Availability Statement:** All data underlying tables and statistics presented in the study have been provided in Supporting Information files. Non-aggregated data cannot be shared publicly because of agreements with Duke University Health System and Durham County Detention Facility. To obtain access to data, researchers would need to contact Nicole Schramm-Sapyta, Maria Tackett, and Michele Easter to join the research team, which would require appropriate research credentials and a relevant, acceptable analysis plan. Following this initial approval, additional approval of the overall plan would be sought from contributing data partners, and finally institutional approvals including sponsorship as Duke affiliate (requiring a background check), and approvals (human subjects protection, Duke's protected computing environment).

**Funding:** Funding was provided as part of an undergraduate course series by Duke University Bass Connections Program (NSS and MT, https://bassconnections.duke.edu/) and Rhodes Information Initiative at Duke Data Plus Program (NSS and MT, https://bigdata.duke.edu/participate/data-plus/). Additional funding for Michele Easter's time was provided by the Wilson Center for Science & Justice, but not specifically for this study, (https://wcsj.law.duke.edu/). The sponsors and funders played no role in study design, data collection and analysis, decision to publish, nor preparation of the manuscript.

**Competing interests:** The authors have declared that no competing interests exist.

## Introduction

Americans have shorter lives and worse health than those living in comparably developed countries [1], while having the highest rate of incarceration in the world [2]. Those who have experienced incarceration often have significant health and healthcare needs and barriers, including a need for primary care that often goes unmet [3–7]. Individuals from minoritized communities, with adverse childhood experiences, or with other disadvantages are more likely to become involved in the criminal legal system for a number of reasons [8–11]. These disadvantages, in combination with sequelae of confinement (e.g., trauma, infectious disease exposure, life disruption, [12]) and reentry challenges make this population particularly vulnerable to health problems and contribute to health inequities [13–15]. Individuals with criminal legal involvement have a higher prevalence of chronic and infectious disease [16–19], creating needs that could be alleviated by a robust connection to primary care.

A key point of opportunity for connection to primary care is the transition out of incarceration, back into the community. With this in mind, it is important to consider the distinction between jail and prison release. Nearly half a million people were released from U.S. federal and state prisons in 2022 [9], yet jail releases are estimated to be over 7 million (7 million admissions at an average stay of 32 days [11]) Jail-involved people, as compared to prison-involved people, are arguably the most relevant to local community healthcare providers, because they are a much larger population and most return quickly to the community; the vast majority are un-convicted and awaiting court action [11].

Those with primary care needs face significant disruption in continuity across carceral and community systems [20–24]. Community-based primary care providers can play a significant role in improving the health and quality of life for this population. Although people in prison and jail are entitled to and may receive primary care during detention and incarceration, they typically face significant challenges in engagement and continuity of care across the transition. These challenges include lack of insurance, fear of discrimination, lack of coordination across systems, and distrust [6, 23, 25, 26]. Releases can also be unpredictable and therefore difficult to coordinate with community services. Specialized interventions exist to overcome these challenges [6, 27, 28] but are not widely implemented nationwide.

Healthcare systems often lack information about how their local county jail population uses their services. This study uses a unique county-wide dataset to follow the entire population of people detained or incarcerated in one urban county detention facility in North Carolina and to characterize their community healthcare use, using electronic health records that capture nearly all primary care visits in the county. Frequent ED use is more common in this population than in the general population [29] and the general patient population of this specific healthcare system [30]. Ideally, people with frequent ED visits and other acute illness care such as inpatient (IP) stays should become engaged in outpatient care and thereby avoid preventable ED/IP. How far is reality from that ideal? This paper provides an overview of community healthcare use by a jail-involved population with a focus on primary care, and compares factors associated with greater engagement for subgroups defined by chronic disease.

## Materials and methods

### 1. Sample and data

Data were collected from a southeastern mid-sized city in a state that had not expanded Medicaid at the time of the study. Every person who was booked into the county detention facility, 2014–2020, was included in the initial study sample (n = 29,120), i.e., anyone serving a sentence, awaiting trial, or released without spending the night. Their interactions with the

healthcare system could have occurred before, during, or after booking. The present analysis includes only the healthcare-using subsample, i.e., those with a 2014–2020 healthcare encounter accompanied by a diagnosis reflecting a provider's clinical impression (n = 17,974/29,120, 61.7% of the initial sample, S1 Table in S1 File).

Retrospective data on bookings and healthcare encounters January 1, 2014-December 31, 2020 from the detention facility and healthcare system included all county Emergency Departments and hospitals except VA, and most primary care clinics in the county, including all clinics known to serve poor and underinsured clients except one. Retrospective data were accessed for research purposes by informaticians at the detention facility and provided to informaticians at the healthcare system for linkage to data from healthcare records; there was no recruitment or other subject contact for this study. Healthcare system informaticians removed all direct identifiers and adjusted all dates to create a set of linked, limited datasets that were shared with the research team on June 16, 2021. The limited datasets were accessible only via the healthcare system's secure computing environment and a user-limited study folder. Authors did not have access to information that could identify individual participants. The Duke University School of Medicine IRB included a prisoner representative, waived the requirement of consent for use of these retrospective records, and approved this study (Pro00102050).

## 2. Measurement

Healthcare encounters were categorized into types (e.g., primary care, Emergency Department (ED), inpatient, urgent care, testing.). Primary care and ED encounters were aggregated into 7-year counts (2014–2020) and calendar-year counts (2014, 2015, etc.). Frequent ED use (FEDU) was defined as 4+ visits in a single calendar year, a commonly used threshold [31–33]. Primary care engagement included visits with any primary care provider and excluded OB/GYN and behavioral health. Encounters were classified as "uninsured" if no payor was listed; "Medicaid/state-funded" (including pending); and "other" if Medicare, private insurance, or another payor was recorded. The first-listed diagnoses at each encounter were assigned to categories according to ICD10 chapters [34]. People were classified into one of three chronic condition statuses: having either (1) a "priority" chronic condition (obstructive lung disease, kidney disease, diabetes, hypertension, heart failure, myocardial infarction, as classified by [35], (2) other chronic condition (as classified by Chronic Condition Indicator Refined [36]), or (3) no record of a chronic condition. The 6 "priority" conditions were chosen because such patients would typically need 2 or more primary care visits annually. In addition, diagnoses were categorized as ambulatory care sensitive conditions [37] and chronic behavioral health conditions (ICD mental disorder chapter excluding tobacco use disorder). Person-level, mutually exclusive categories were created for insurance (ever Medicaid/state insurance, ever other insurance and never Medicaid/state insurance, and never insurance) and chronic condition status (ever "priority" chronic condition, ever other chronic condition, and no chronic condition). Demographic characteristics as recorded by the detention facility were collapsed to facilitate analysis: sex/gender (male, female/other); race (Black, white, other); and ethnicity (Latino, not Latino). Age was measured as of 2021 and categorized into age bands (18–29, 30–39, 40–49, 50+). Bookings and incarceration days were totaled over the 7-year observation period, and bookings categorized as (1, 2, 3–4, and 5 or more bookings).

## 3. Analysis

Descriptive statistics were used to characterize healthcare utilization and diagnoses. Multivariable logistic regression models were used to estimate the odds of receiving any primary care

over the 7-year period for the full sample as well as for three subgroups: those with one or more of the 6 chronic conditions, those with any other chronic condition, and those with no observed chronic condition. Independent variables were gender, race, ethnicity, age, number of jail bookings, insurance type (ever observed), and total incarceration days (to adjust for time out of community). Zero-inflated negative binomial (ZINB) regression models were used to estimate associations between independent variables and the number of primary care visits over 7 years ("count model"), and the "excess zeros", i.e., the absence of any primary care visits ("zero model"), which could be shaped by different factors than the number of visits. Formal one-sided tests of overdispersion using the Pearson statistic supported use of ZINB. These ZINB models were estimated for the same four groups as the logistic regression and count models included the same independent variables as the logistic regression models for all four groups, as did the zero model for the full set of patients. In zero models for subgroups defined by chronic conditions, several independent variables were collapsed (age from 4 categories to 2) or omitted (Latino, insurance) due to lack of convergence and inflated estimates that were not interpretable for one or more groups. SAS 9.4 was used for all analyses; PROC GENMOD was used to carry out logistic and ZINB models [38].

## Results

### 1. Use of primary care and other healthcare by one county's jail-involved population

About half (50.6%) of the sample had a primary care visit over the 7-year study period (Table 1, left panel). Consistent with earlier research (Authors 2023), the most common type of encounter with the healthcare system was an ED visit; most (84.1%) had one, half staying overnight at least once. Far fewer visited urgent care (37.3%) or were admitted for inpatient hospitalization (16.1%) during that time period.

**Table 1. Healthcare encounters 2014–2020 by people with past/future detention who had contact with the health system (n = 17,974 people).**

| | Ever had this encounter type (n = 17,974) | | Number of encounters for those with any of this type | | | | |
|---|---|---|---|---|---|---|---|
| | n | % | Mean | SD | Mdn. | Range | |
| Encounter type (ever) | | | | | | | |
| Primary care | 9,093 | 50.6 | 8.5 | (11.7) | 4 | 1 - | 173 |
| Emergency Dept. | 15,109 | 84.1 | 7.2 | (11.8) | 4 | 1 - | 309 |
| ED overnight* | 9,373 | 52.2 | 3.4 | (5.7) | 2 | 1 - | 183 |
| Nights total (over 7 years) | | | 8.6 | (21.6) | 2 | 1 - | 501 |
| Inpatient admission** | 2,884 | 16.1 | 1.7 | (1.3) | 1 | 1 - | 14 |
| Urgent care | 6,696 | 37.3 | 3.3 | (3.8) | 2 | 1 - | 46 |
| Testing | 13,027 | 72.5 | 8.6 | (12.8) | 4 | 1 - | 191 |
| Psych/Behavioral | 1,780 | 9.9 | 7.4 | (16.5) | 2 | 1 - | 235 |
| Payer (ever) | | | | | | | |
| Uninsured | 14,947 | 83.2 | 11.1 | (18.3) | 5 | 1 - | 400 |
| Medicaid/state-insured | 7,353 | 40.9 | 30.6 | (45.0) | 15 | 1 - | 608 |
| Other | 8,519 | 47.4 | 14.5 | (28.8) | 5 | 1 - | 639 |
| Corrections | 641 | 3.6 | 4.5 | (6.0) | 3 | 1 - | 59 |

*Any ED encounter where admission date ~ = discharge date

**Hospital encounter that is not an ED visit and had an overnight

Of those who had a primary care visit, the mean number of visits was 8.5 over the 7-year period (11.7 SD), with a range 1–173 (Table 1, right panel) and a median of 4 visits. For those with an ED visit, the mean number of ED visits was only slightly lower at 7.2 (SD 11.8), and ranged 1–309, and the median was also 4. For those who ever stayed overnight at the ED. Those who were hospitalized had a mean of 1.7 admissions, with a median of 1 and range 1–14. Some individuals had more than 800 encounters with the healthcare system over 7 years (S2 Table in S1 File).

With regard to insurance (non-mutually exclusive categories), most (83.2%) of the sample had at least one uninsured encounter over the 7-year period. About 41% ever had an encounter paid for by Medicaid or state fund, and nearly half ever had an encounter paid for by another type of payer, including private insurance, Medicare, a carceral facility, or another type of payer. Those with Medicaid or state-funded health care had more visits (mean 30.6 visits) than those who were uninsured (mean 11.1 visits) or had other types of coverage (mean 14.5 visits).

## 2. Characteristics of healthcare user types: demographic, detention-related, and clinical features

The sample was classified into four groups based on frequency and type of healthcare utilization (Table 2): "Any IP" ever had an inpatient admission; "Frequent ED" ever had a year of frequent ED use (and never an inpatient admission), "Any ED" ever used the ED (but never frequently and never inpatient), and "No ED/IP" never used ED or inpatient. The largest group ("Any ED" 54.2%) visited the ED at least once, but never frequently, and were never hospitalized. The other three groups were approximately equal in size (15–16%): those who never went to the ED nor hospital ("No ED/IP"); those who went to the ED frequently but were never hospitalized ("Frequent ED"); and those who were hospitalized ("Any IP"). Of the "Any IP" group, 42% were also frequent ED users at some point over the study period. All demographic, detention-related, and clinical characteristics were statistically significantly associated with healthcare user type.

The "Frequent ED" group was the only group in which most people (61.5%) had more than one jail booking (combining categories), and 25.2% had 5 or more bookings, compared to 5.6–15.4% for the other groups. The "Frequent ED" group also had the highest percentage of Black individuals (71.8%), the lowest percentage of white individuals (17.6%) and more older individuals, with 28.1% 50 or older (compared to 20.2–22.6% for the other groups) The "Frequent ED" group and the "Any IP" group had the highest proportions with one or more of the 6 "priority" chronic conditions (54.6% and 50.2%), ambulatory care sensitive conditions (63.1% and 64.6%), and behavioral health conditions (64.3% and 70.7%).

By contrast, the "No ED/IP" group had the fewest jail bookings; 72.9% had one jail booking only, the minimum to be included in the study (vs. 38.5–54.9% of other groups). The "No ED/IP" group also had highest percentage of white individuals (32.4% vs. 17.6–21.2%) and the lowest of Black (45.7% vs. 63.4–71.8%), as well as the highest % Latino (7.9% vs. 1.9–4.9%). This group also had the lowest proportion ever having public insurance and the highest proportion ever having a private or other non-public payer. The "No ED/IP" group had the fewest diagnosed chronic conditions, ambulatory care sensitive conditions, and behavioral health conditions.

Of the remaining two groups, the "Any IP" group had the highest proportion female (63.5% vs. 22.0–31.7% for other groups), and the highest proportion ever having public insurance (79.0% vs. 17.1–57.3%). Characteristics of the "Any ED" group usually fell between or were similar to those of other groups.

**Table 2. Demographic, detention-related, and clinical characteristics by healthcare-user type (n = 17,974 people).**

| | No ED/IP (n = 2,696) | | ED but never FF (n = 9,745) | | ED FF (n = 2,649) | | IP (n = 2,884) | | All groups (n = 17,974) | |
|---|---|---|---|---|---|---|---|---|---|---|
| | n | % | n | % | n | % | n | % | n | % |
| **Sex** | | | | | | | | | | |
| Male | 2,057 | (76.3) | 7,600 | (78.0) | 1,810 | (68.3) | 1,051 | (36.4) | 12,518 | (69.7) |
| Female | 639 | (23.7) | 2,145 | (22.0) | 839 | (31.7) | 1,833 | (63.6) | 5,456 | (30.4) |
| **Race/ancestry** | | | | | | | | | | |
| White | 874 | (32.4) | 2,018 | (20.7) | 467 | (17.6) | 611 | (21.2) | 3,970 | (22.1) |
| Black | 1,232 | (45.7) | 6,178 | (63.4) | 1,903 | (71.8) | 1,855 | (64.3) | 11,168 | (62.1) |
| Other | 590 | (21.9) | 1,549 | (15.9) | 279 | (10.5) | 418 | (14.5) | 2,836 | (15.8) |
| **Ethnicity** | | | | | | | | | | |
| Not Latino | 2,484 | (92.1) | 9,267 | (95.1) | 2,599 | (98.1) | 2,799 | (97.1) | 17,149 | (95.4) |
| Latino | 212 | (7.9) | 478 | (4.9) | 50 | (1.9) | 85 | (3.0) | 825 | (4.6) |
| **Age** | | | | | | | | | | |
| 18–29 | 690 | (25.6) | 2,635 | (27.0) | 489 | (18.5) | 841 | (29.2) | 4,655 | (25.9) |
| 30–39 | 828 | (30.7) | 3,158 | (32.4) | 831 | (31.4) | 959 | (33.3) | 5,776 | (32.1) |
| 40–49 | 570 | (21.1) | 1,987 | (20.4) | 584 | (22.1) | 448 | (15.5) | 3,589 | (20.0) |
| 50+ | 608 | (22.6) | 1,965 | (20.2) | 745 | (28.1) | 636 | (22.1) | 3,954 | (22.0) |
| **Number of bookings** | | | | | | | | | | |
| 1 booking | 1,965 | (72.9) | 5,089 | (52.2) | 1,021 | (38.5) | 1,582 | (54.9) | 9,657 | (53.7) |
| 2 bookings | 390 | (14.5) | 1,725 | (17.7) | 483 | (18.2) | 507 | (17.6) | 3,105 | (17.3) |
| 3–4 bookings | 190 | (7.1) | 1,431 | (14.7) | 477 | (18.0) | 412 | (14.3) | 2,510 | (14.0) |
| 5+ bookings | 151 | (5.6) | 1,500 | (15.4) | 668 | (25.2) | 383 | (13.3) | 2,702 | (15.0) |
| **Insurance (over 7 years)** | | | | | | | | | | |
| Never had payer | 756 | (28.0) | 3,438 | (35.3) | 531 | (20.1) | 103 | (3.6) | 4,828 | (26.9) |
| Ever had public payer | 460 | (17.1) | 3,097 | (31.8) | 1,519 | (57.3) | 2,277 | (79.0) | 7,353 | (40.9) |
| Other | 1,480 | (54.9) | 3,210 | (32.9) | 599 | (22.6) | 504 | (17.5) | 5,793 | (32.2) |
| **Diagnoses (ever)** | | | | | | | | | | |
| **Ever chronic condition** | | | | | | | | | | |
| Any of 6 chronic conditions* | 546 | (20.3) | 2,723 | (27.9) | 1,446 | (54.6) | 1,447 | (50.2) | 6,162 | (34.3) |
| Any other chronic condition | 1,013 | (37.6) | 5,101 | (52.3) | 1,159 | (43.8) | 1,393 | (48.3) | 8,666 | (48.2) |
| No chronic condition | 1,137 | (42.2) | 1,921 | (19.7) | 44 | (1.7) | 44 | (1.5) | 3,146 | (17.5) |
| **Ever ambulatory care sensitive condition (PQI any of 10)** | | | | | | | | | | |
| No | 2,163 | (80.2) | 6,819 | (70.0) | 978 | (36.9) | 1,022 | (35.4) | 10,982 | (61.1) |
| Yes | 533 | (19.8) | 2,926 | (30.0) | 1,671 | (63.1) | 1,862 | (64.6) | 6,992 | (38.9) |
| **Ever mental, substance, or alcohol disorder (excl. tobacco)** | | | | | | | | | | |
| No | 2,173 | (80.6) | 6,584 | (67.6) | 946 | (35.7) | 844 | (29.3) | 10,547 | (58.7) |
| Yes | 523 | (19.4) | 3,161 | (32.4) | 1,703 | (64.3) | 2,040 | (70.7) | 7,427 | (41.3) |

All crosstabulations were statistically significant at p<0.001

*One or more of the following chronic conditions: obstructive lung disease, kidney disease, diabetes, hypertension, heart failure, myocardial infarction

With regard to chronic conditions, most of the sample had a chronic condition, either one or more of the 6 "priority" conditions (34.3%) or another chronic condition (48.2%); 17.5% had no record of a chronic condition. Virtually all people in the "Frequent ED" and "Ever IP" groups were ever diagnosed with a chronic condition, and half or more had one of the "priority" conditions (54.6% and 50.2% respectively). Most people in "Frequent ED" and "Ever IP" had an ambulatory care sensitive condition (63.1% or 64.6% respectively), and/or a chronic behavioral health condition (64.3% and 70.7% respectively).

### 3. Common diagnoses at ED or IP by type of healthcare user

To better understand the health needs of healthcare user types, we classified all diagnoses received at ED or IP for the three groups that used these services ("Any ED", "Frequent ED", "Any IP"). The two ED groups had the same five diagnosis types among their top 5 (at ED): Injury/poisoning, mental/behavioral, musculoskeletal/connective, digestive, respiratory. Mental/behavioral diagnoses were most common among the "Frequent ED" group, and injury/poisoning diagnoses (which includes drug or alcohol intoxication or overdose) among the "Any ED" group. For the "Any IP" group, mental/behavioral diagnoses were the most common, but the next 4 most common diagnoses were not shared with the other two groups: pregnancy/childbirth, metabolic diseases, neoplasms, and nervous system disorders.

### 4. Primary care visits by healthcare user groups and chronic condition groups

Out of a total of 514,725 encounters (S3 Table in S1 File), 15.0% (n = 77,224) were for primary care. About a third of all visits were uninsured (32.3% of all encounters, 32.6% of primary care visits), and about 40% were covered by Medicaid or state insurance (43.7% of all encounters, 39.8% of primary care visits). The remaining quarter were covered by another insurance plan (24.0% of all encounters, 28.6% of primary care visits).

The four healthcare user groups varied in their use of primary care (Table 3, upper panel). The "Any IP" group used primary care the most, with the highest proportion ever using it (69.4%) and highest mean visits (12.1). The "Frequent ED" group had the next highest use, with 62.6% ever using it and a mean of 10.5 visits for those who did. Consistency of use, measured as years with at least one primary care visit, was also highest for "Any IP" followed by the "Frequent ED" group (S4 Table in S1 File).

Because people with "Any IP" or "Frequent ED" use patterns are likely to have needs that would benefit from greater engagement in primary care, we compared groups by whether they ever had 2 or more primary care visits in any calendar year. The "Any IP" group had the highest engagement, with 42.6% ever having 2+ visits, followed by the "Frequent ED" group, with 35.0% having 2+ visits (consistency of use followed the same pattern).

**Table 3. Healthcare-users with past/future detention and their primary care engagement 2014–2020 by healthcare user type and chronic disease status (n = 17,974).**

| | n | % | Primary care (over 7 years) | | | | | Years with 2+ primary care visits (over 7 years) | | | | |
| | | | Any visit | | # Visits (if any) | | | Any year | | # Years (if any) | | |
| | n | % | n | % | Mean | SD | Mdn. | n | % | Mean | SD | Mdn. |
|---|---|---|---|---|---|---|---|---|---|---|---|---|
| *Types of healthcare-user* | | | | | | | | | | | | |
| No ED or IP visits | 2,696 | 15.0 | 1,401 | 52.0 | 5.4 | (6.7) | 3 | 526 | 19.5 | 1.9 | (1.4) | 1 |
| Any ED (not frequent, no IP) | 9,745 | 54.2 | 4,032 | 41.4 | 7.0 | (9.4) | 3 | 1,778 | 18.3 | 2.2 | (1.5) | 2 |
| Frequent ED (no IP) | 2,649 | 14.7 | 1,659 | 62.6 | 10.5 | (13.9) | 5 | 928 | 35.0 | 2.5 | (1.7) | 2 |
| Any IP (may also have frequent ED*) | 2,884 | 16.1 | 2,001 | 69.4 | 12.1 | (15.2) | 7 | 1,227 | 42.6 | 2.6 | (1.7) | 2 |
| *Chronic condition status (ever)* | | | | | | | | | | | | |
| Selected chronic condition** | 6,162 | 34.3 | 4,502 | 73.1 | 12.2 | (14.4) | 8 | 2,909 | 47.2 | 2.6 | (1.7) | 2 |
| Other chronic condition | 8,666 | 48.2 | 3,949 | 45.6 | 5.3 | (7.0) | 3 | 1,475 | 17.0 | 1.8 | (1.3) | 1 |
| No chronic condition | 3,146 | 17.5 | 642 | 20.4 | 2.0 | (1.9) | 1 | 75 | 2.4 | 1.1 | (0.4) | 1 |
| *All groups* | *17,974* | *100.0* | *9,093* | *50.6* | *8.5* | *(11.7)* | *4* | *4,459* | *24.8* | *2.3* | *(1.6)* | *2* |

ED = Emergency Department; IP = Inpatient admission; Frequent ED = any calendar year with 4 or more ED visits

*42.2% (n = 1,216) of IP group had at least one calendar year of frequent ED visits.

**One or more of the following chronic conditions: obstructive lung disease, kidney disease, diabetes, hypertension, heart failure, myocardial infarction

We also compared use of primary care by chronic condition status Those with "priority" conditions were more connected to primary care than the other two groups: 73.1% ever had a primary care visit (vs. 45.6% and 20.4%); the highest number of visits (mean of 12.2 vs. 5.3 and 2.0), and more years with 2+ primary care visits (2.6 vs. 1.8 and 1.1).

## 5. Correlates of primary care engagement

What factors were associated with primary care engagement? Demographic characteristics, number of bookings, and insurance factors were considered together in models predicting any primary care visit over 7 years (Table 4 upper panel: logistic regression models) and the number of primary care visits over 7 years (Table 4 lower panel: zero-inflated negative binomial regression count model, see S5 Table in S1 File for zero model). In addition to the full sample, separate models were estimated for three subsamples: people with "priority" chronic conditions, other chronic conditions, and no recorded chronic conditions.

In general, across most models and chronic disease subgroups, being female, Latino, older, or insured was associated with having primary care. Insurance (primarily Medicaid/state insurance) and age (primarily age 50+) had the largest effect sizes for people with chronic conditions. However, for the subgroup with no chronic conditions, being Latino was associated with the largest effect sizes (ever having access and having more visits), more so than age, and being female was associated with a lower odds of ever having primary care. Being Black and having a higher number of bookings each had mixed results across models and are described separately.

In full group models (logistic, count, and zero), the largest effect sizes were associated with having public insurance compared to no payer listed (a 4.9 times greater odds of ever having primary care and an average of 3.6 additional visits) and age, particularly being 50+ years old compared to 18–29 (e.g., a 4.0 times greater odds of ever having a visit, and an average of 4.0 additional encounters). In the zero model for the full group, being Latino and ever having "other" insurance were additionally among the largest effect sizes, in the direction of greater primary care engagement.

Similar results were found in the two subsamples with chronic conditions; the largest effect sizes were also with older age, Medicaid/state insurance, and other insurance in both logistic and count models. In the model for the subsample with no chronic conditions, the largest effect sizes were associated with public insurance, other insurance, and being Latino.

Less consistency was seen across models in estimates related to being Black and the number of jail bookings. Compared to being white, being Black was associated with a greater odds of ever having primary care for those with "priority" chronic conditions, but a lower odds for those with other chronic conditions. Being Black was also associated with fewer visits in the count models for the full group and both subsamples with chronic conditions.

The number of jail bookings had an impact on the number of primary care encounters but mixed results for ever having access. Regarding one-time access, having more jail bookings usually was not statistically significantly associated. By contrast, in count models, having a high number of jail bookings was consistently associated with fewer visits. This pattern was seen in the full group. (3–4 bookings and 5+ bookings vs. 1), the subgroup with "priority" chronic conditions (same) and those with other chronic conditions (5+ bookings vs. 1). Specifically, for those with the priority chronic diseases, people with 3–4 bookings (or 5+ bookings) have an average of 80% as many visits compared to those with only one booking, i.e., a 20% decrease in visits. These associations with bookings were independent of total time detained in jail.

**Table 4. Characteristics associated with primary care engagement among people with jail involvement: Logistic regression and zero-inflated negative binomial count models for full sample and subgroups based on chronic condition history (n = 17,974 people).**

### A. Logistic regression models predicting any primary care encounter over 7 years

| | All n = 17,974 | | | One or more of 6 chronic conditions (n = 6,162) | | | One or more other chronic condition(s) (n = 8,666) | | | No chronic conditions (n = 3,146) | | |
|---|---|---|---|---|---|---|---|---|---|---|---|---|
| | O.R. | 95% C.I. | p | O.R. | 95% C.I. | p | O.R. | 95% C.I. | p | O.R. | 95% C.I. | p |
| Gender (ref = male) | | | | | | | | | | | | |
| Female | 1.4 | (1.3- 1.5) | <0.001 | 1.6 | (1.4- 1.9) | <0.001 | 1.4 | (1.3- 1.6) | <0.001 | 0.8 | (0.6- 1.0) | 0.029 |
| Race (ref = white/other) | | | | | | | | | | | | |
| Black | 1.0 | (0.9- 1.1) | 0.948 | 1.2 | (1.0- 1.3) | 0.029 | 0.9 | (0.8- 1.0) | 0.0454 | 0.9 | (0.7- 1.1) | 0.174 |
| Ethnicity (ref = not Latino) | | | | | | | | | | | | |
| Latino | 1.7 | (1.4- 2.0) | <0.001 | 1.7 | (1.2- 2.5) | 0.007 | 1.7 | (1.4- 2.2) | <0.001 | 2.1 | (1.6- 2.9) | <0.001 |
| Age (ref = 18–29) | | | | | | | | | | | | |
| 30–39 | 1.6 | (1.5- 1.8) | <0.001 | 2.3 | (1.9- 2.8) | <0.001 | 1.3 | (1.2- 1.4) | <0.001 | 1.3 | (1.0- 1.6) | 0.022 |
| 40–49 | 2.7 | (2.4- 2.9) | <0.001 | 3.6 | (3.0- 4.4) | <0.001 | 1.7 | (1.5- 2.0) | <0.001 | 1.5 | (1.2- 2.0) | 0.002 |
| 50 or older | 4.0 | (3.6- 4.4) | <0.001 | 4.4 | (3.7- 5.2) | <0.001 | 2.0 | (1.7- 2.3) | <0.001 | 1.3 | (1.0- 1.8) | 0.091 |
| Number of bookings over 7 years (ref = 1 booking) | | | | | | | | | | | | |
| 2 bookings | 1.1 | (1.0- 1.2) | 0.118 | 1.1 | (0.9- 1.3) | 0.275 | 0.9 | (0.8- 1.1) | 0.3789 | 1.3 | (1.0- 1.7) | 0.031 |
| 3 or 4 bookings | 1.0 | (0.9- 1.1) | 0.505 | 0.9 | (0.8- 1.1) | 0.329 | 1.0 | (0.9- 1.1) | 0.901 | 1.0 | (0.8- 1.4) | 0.825 |
| 5 or more bookings | 1.0 | (0.9- 1.2) | 0.427 | 1.0 | (0.9- 1.3) | 0.669 | 0.9 | (0.8- 1.0) | 0.1408 | 0.8 | (0.6- 1.2) | 0.363 |
| Insurance over 7 years (ref = Never had payer) | | | | | | | | | | | | |
| Ever Medicaid/state | 4.9 | (4.5- 5.3) | <0.001 | 3.4 | (2.9- 4.1) | <0.001 | 3.5 | (3.1- 4.0) | <0.001 | 2.2 | (1.7- 2.9) | <0.001 |
| Other payer | 2.6 | (2.4- 2.9) | <0.001 | 1.9 | (1.6- 2.3) | <0.001 | 2.6 | (2.3- 2.9) | <0.001 | 1.8 | (1.5- 2.2) | <0.001 |
| Total incarceration days over 7 years | 1.0 | (1.0- 1.0) | <0.001 | 1.0 | (1.0- 1.0) | 0.032 | 1.0 | (1.0- 1.0) | <0.001 | 1.0 | (1.0- 1.0) | 0.037 |

### B. Zero-inflated negative binomial regression model predicting count of primary care encounters over 7 years (n = 17,974 people)

| Count model* | All n = 17,974 | | | One or more of 6 chronic conditions (n = 6,162) | | | One or more other chronic condition(s) (n = 8,666) | | | No chronic conditions (n = 3,146) | | |
|---|---|---|---|---|---|---|---|---|---|---|---|---|
| | Exp. Est. ** | 95% C.I. | p | Exp. Est. ** | 95% C.I. | p | Exp. Est. ** | 95% C.I. | p | Exp. Est. ** | 95% C.I. | p |
| Gender (ref = male) | | | | | | | | | | | | |
| Female | 1.7 | (1.6- 1.8) | <0.001 | 1.4 | (1.3- 1.6) | <0.001 | 1.9 | (1.8- 2.2) | <0.001 | 0.9 | (0.6- 1.2) | 0.388 |
| Race (ref = white/other) | | | | | | | | | | | | |
| Black | 0.9 | (0.9- 1.0) | 0.007 | 0.9 | (0.9- 1.0) | 0.046 | 0.8 | (0.7- 0.9) | <0.001 | 1.0 | (0.8- 1.3) | 0.926 |
| Ethnicity (ref = not Latino) | | | | | | | | | | | | |
| Latino | 0.9 | (0.8- 1.0) | 0.194 | 1.2 | (1.0- 1.5) | 0.073 | 1.3 | (1.1- 1.6) | 0.0118 | 2.0 | (1.5- 2.7) | <0.001 |
| Age (ref = 18–29) | | | | | | | | | | | | |
| 30–39 | 1.6 | (1.5- 1.7) | <0.001 | 1.9 | (1.7- 2.1) | <0.001 | 1.3 | (1.2- 1.4) | <0.001 | 1.0 | (0.8- 1.3) | 0.932 |
| 40–49 | 2.5 | (2.3- 2.7) | <0.001 | 2.6 | (2.3- 2.9) | <0.001 | 1.6 | (1.4- 1.8) | <0.001 | 1.1 | (0.8- 1.5) | 0.735 |
| 50 or older | 4.0 | (3.6- 4.3) | <0.001 | 3.4 | (3.0- 3.8) | <0.001 | 2.2 | (1.9- 2.6) | <0.001 | 1.0 | (0.7- 1.4) | 0.793 |
| Number of bookings over 7 years (ref = 1 booking) | | | | | | | | | | | | |
| 2 bookings | 1.0 | (0.9- 1.0) | 0.376 | 1.0 | (0.9- 1.0) | 0.276 | 0.9 | (0.8- 1.0) | 0.1097 | 1.0 | (0.7- 1.3) | 0.76 |
| 3 or 4 bookings | 0.9 | (0.8- 0.9) | 0.001 | 0.8 | (0.8- 0.9) | 0.002 | 0.9 | (0.8- 1.0) | 0.1534 | 0.8 | (0.5- 1.2) | 0.256 |
| 5 or more bookings | 0.8 | (0.7- 0.9) | <0.001 | 0.8 | (0.7- 0.9) | <0.001 | 0.7 | (0.6- 0.8) | <0.001 | 1.3 | (0.8- 2.1) | 0.333 |
| Insurance over 7 years (ref = Never had payer) | | | | | | | | | | | | |
| Ever Medicaid/state | 3.6 | (3.3- 4.0) | <0.001 | 3.2 | (2.8- 3.5) | <0.001 | 4.3 | (3.8- 4.8) | <0.001 | 2.7 | (2.0- 3.6) | <0.001 |
| Other payer | 2.2 | (2.0- 2.4) | <0.001 | 1.9 | (1.7- 2.1) | <0.001 | 3.2 | (2.8- 3.6) | <0.001 | 2.0 | (1.7- 2.5) | <0.001 |

(*Continued*)

**Table 4.** (Continued)

| A. Logistic regression models predicting any primary care encounter over 7 years | | | | | | | | | | | | |
|---|---|---|---|---|---|---|---|---|---|---|---|---|
| | All n = 17,974 | | | One or more of 6 chronic conditions (n = 6,162) | | | One or more other chronic condition(s) (n = 8,666) | | | No chronic conditions (n = 3,146) | | |
| | O.R. | 95% C.I. | | p | O.R. | 95% C.I. | | p | O.R. | 95% C.I. | | p | O.R. | 95% C.I. | | p |
| Total incarceration days over 7 years | 1.0 | (1.0- | 1.0) | <0.001 | 1.0 | (1.0- | 1.0) | <0.001 | 1.0 | (1.0- | 1.0) | 0.2548 | 1.0 | (1.0- | 1.0) | 0.02 |

*Zero model shown in S5 Table in S1 File

**Exponentiated estimate: the change in the expected (mean) count for one unit increase in the predictor.

## Discussion

This study found that about half of people who were detained or incarcerated in one county jail 2014–2020 and interacted with the local healthcare system had seen a primary care provider during that time period. Rates were higher for those whose healthcare use or diagnoses indicated greater need, such as those with inpatient stays, frequent ED use, and diagnosis of one or more "priority" chronic conditions. Typically, these individuals still underutilized primary care for important health conditions, with only a third or fewer of those groups having a visit most years and under half ever having a year with two primary care visits, the recommended minimum for people with the priority diagnoses. In 2019, 67% of Americans 18 or older had a primary care-like visit in the past year (i.e., a wellness visit, physical, or check-up was had by 78.9% of the 84.9% who saw a physician [39]).

This jail-involved sample was enriched with characteristics associated with lower engagement in care; in the general U.S. population, non-White, younger, male, or uninsured individuals tend to have lower access to a "usual source of care" compared to White, older, female, or insured individuals [40]. More than three quarters of the sample (83%) ever had a healthcare encounter that was uninsured, and when viewed across 7 years, the modal category had received Medicaid/state insurance (40.9%), and the next largest group had never had a payer listed (26.9%).

Women, who are underrepresented in the jail population, were overrepresented among inpatients compared to men. This was consistent with pregnancy/childbirth being one of the top 5 diagnostic families for inpatient encounters but not in the other healthcare user groups, and lower rates of primary care engagement. Women of reproductive age may get Medicaid when pregnant and be admitted for childbirth. Their involvement in the healthcare system may have been driven by healthy childbearing, and their primary care needs may have been minimal or addressed well by OBGYN specialists at the time of care.

There was evidence of feedback loop between insurance and health service type. Nearly 80% of the inpatient group had a public payer, likely due to the hospital assisting eligible pregnant women and others to get care covered by Medicaid or other local resources. There may also be selection effects in which those who cannot pay are referred elsewhere for non-emergent care, e.g., a state psychiatric hospital rather than a private academic hospital. Only 3.6% of the inpatient group never had a payer listed.

The frequent ED-using subgroup had significant health and other challenges: a quarter had 5 or more bookings during the study period, more than half had at least one of the 6 priority chronic diseases, nearly two thirds ever had an ambulatory care sensitive condition, and similar numbers ever had a behavioral health disorder. The inpatient group was similar in these respects to the frequent ED group with the notable exception of having fewer jail bookings. In this sample, inpatients had better access and engagement with primary care than frequent ED-users despite similar diagnoses. This difference is consistent with system-level factors, such as

requirements surrounding discharge and referral for inpatients [41], as well as person-level factors that may be associated with both frequent ED-use and multiple jail bookings.

Results showed that jail bookings were associated with a reduced number of visits, though not one-time access. This finding was driven by the two subgroups with chronic disease, who most need regular primary care. Having more jail bookings was independently associated with reduced repeat primary care visits, even controlling for age, insurance, other demographic characteristics, and the number of days detained. This suggests that much more needs to be done for this vulnerable group to promote and support ongoing engagement in primary care; a person may go once but to have a real impact there should be repeated visits.

A similar pattern of associations was found for being Black as for jail bookings; being Black was associated with reduced visits but not one-time access, suggesting a similar need for support to sustain engagement. Being Black was independently associated with reduced visits even when adjusted for insurance, number of bookings, time incarcerated and other characteristics. Sustained engagement in primary care is likely shaped by structural and institutional racism, and likely to be worse for Black men [42, 43].

More robust linkage efforts following ED and IP are needed, especially for frequent users. Healthcare systems may provide appointments or referrals to primary care or other providers, but some people lack transportation and also need help to remember appointments; communicate by phone, email, or patient portal; navigate and coordinate across systems; pay for care; and overcome distrust, stigma, and other barriers [23, 44]. One successful intervention model, the Transitions Clinic Network, integrates community health workers with lived experience of incarceration to serve as a liaison between patient and provider, provide person-centered case management to address needs of the patient beyond healthcare, such as housing [24]. An RCT of TCN compared to standard primary care found the program reduced overall emergency room visits by 51% [45]. A propensity-matched comparison also found that TCN reduced preventable hospitalizations, shortened hospitalizations, and reduced incarceration days [46] and costs [47].

Adopting the principles used by the Transitions Clinic Network would reduce barriers to community primary care for jail- and prison-involved individuals. One such program, NC Formerly Incarcerated Transition Program (FIT) [48], is operating in the region and their patients and FIT-supported encounters would be among those in this dataset. FIT's eligibility criteria are recent release from incarceration (prison or jail) and a chronic disease diagnosis (including mental illness and/or substance use disorder), which in this sample would certainly include 6,162 people with one of the priority chronic conditions and many others as well. Considering only those with the six priority conditions, 26.9% never had primary care, which would equate to 1,657 who could use FIT services. But even those who have had primary care also show signs of not being well connected, with 67% (n = 4,128) going most years without a primary care visit, and 52.8% (n = 3,253) never having a year with 2 or more visits, which would be indicated for people with these high-priority chronic diseases. FIT's capacity is primarily defined by the number of community health workers it can employ to manage a caseload of 30–60 clients each. Even taking care of just the 1,657 with serious chronic disease and no evidence of primary care would require at least 27 community health workers, fewer as people stabilize.

There are also specialized models for people with serious mental illness, such as Assertive Community Treatment (ACT), that could support greater engagement in primary care for those with serious mental illness who are eligible [49]. Collaborative care models wherein patients with mild or moderate psychiatric conditions are managed by a consultation team of care manager and consulting psychiatrist are associated with lower readmission rates to IP and ED, and a lower costs of care overall [50], and may use principles from ACT to engage and

redirect frequent users toward other community resources, including outpatient primary care [51]. Initiatives to reduce frequent use of 911, jail, and other city and county resources [52, 53] should include partnerships with primary care providers, along with community paramedics and others.

Greater investment and prioritization to meet the needs of jail-involved frequent ED users with highly complex care needs would benefit patients and providers alike. Healthcare policy makers have argued that improving the U.S. healthcare system requires maximizing the Triple Aim, namely enhancing patient experience, improving population health, and reducing costs, simultaneously [54, 55]. Not only would shifting healthcare use by this patient population from preventable ED/IP to primary care support the Triple Aim, it would also address the additional need articulated in the Quadruple Aim [56]: improving the work life of health care providers, including clinicians and staff. People with preventable conditions or social needs contribute to overcrowding and longer wait times at the ED. There is a subset of jail-involved people who exhibit violent behavior at the ED, whether due to substance or alcohol use disorders, mental health disorders, difficulty communicating, distrust and fear, or frustration by their needs that exceed ED treatment capacity. Healthcare staff who experience violence and/ or are not able to give patients what they need to be healthy and stable may experience physical injury, moral injury, and associated stress and burnout. Interventions such as medical-legal partnerships that effectively address the complex needs of specific populations can improve provider wellbeing because there is some way to help their patients. Further, prioritizing needs of CL-involved people, who are disproportionately from minoritized groups, would support the Quintuple Aim that includes racial equity [57].

Study limitations include measurement of disease based on encounters rather than universal screening; people who come for care are more likely to be diagnosed, particularly if they come frequently. In addition, this analysis excludes those diagnosed by healthcare providers outside the system. Because most people encountered the health system by going to the E.D. and only half ever had a primary care visit, there were nevertheless many opportunities to be diagnosed outside of a primary care context and our approach was able to detect many with chronic disease apart from the primary care context. Moreover, the medical records capture almost all primary care providers likely to serve this county population. Behavioral health-related enhanced services (such as ACT teams) and some routine visits are not as well captured. Our measures of primary care may or may not represent access to a usual source of care; individuals with apparently good engagement actually may be visiting different primary care providers and lack a stable long-term connection to care. This dataset does not support a direct comparison between those who are and are not jail-involved. Last, the data are from just one geographic area. However, this is a payer-agnostic dataset that includes uninsured people, and thus trades geographic coverage for comprehensive inclusion of everyone detained at the county detention facility and nearly everyone who would have sought care nearby.

## Conclusions

In a single county population of individuals ever booked into the detention center, utilization of primary care was inconsistent and overall low. Individuals too often relied on the Emergency Department for care. To promote primary care utilization among individuals who have the combined challenges of repeated jail involvement and chronic disease, it is crucial to focus on engagement, as much as access. This study demonstrated that among jail-involved individuals with high-priority chronic diseases such as hypertension, obstructive lung disease, and diabetes, having repeated bookings was associated with less ongoing engagement in primary care—not one-time access, but repeated, sustained engagement. This barrier was in the context

of low primary care participation for the group as a whole, with under three quarters of those with serious chronic diseases ever seeing a primary care provider and less than half ever having a year with more than one visit. Access to health insurance alone will not resolve the problem; Medicaid expansion should be coupled with specialized, tailored support to promote engagement in primary care. Programs such as Transitions Clinic Network can address barriers to accessing primary care thereby addressing acute and chronic physical health conditions and coordinating or providing mental health and substance abuse care. There should be sufficient support to expand such programs so they can reach every jail-detained individual with chronic disease, along with others who need but do not engage in primary care.

## Supporting information

**S1 File.**
(XLSX)

## Acknowledgments

This work could not have been completed without the cooperation of Sheriff Clarence Birkhead, Major Elijah Bazemore, and Caroline Andrews of the Durham County Sheriff's Office; Gudrun Parmer of the Durham County Criminal Justice Resource Center; Roshanna Parker of the Durham County Justice Services Department; and members of Durham County Stepping Up Initiative. Funding to support data collection and analysis was provided by Duke University Bass Connections Program, Rhodes Information Initiative at Duke Data Plus Program, and the Wilson Center for Science & Justice.

## Author Contributions

**Conceptualization:** Michele M. Easter, Marvin S. Swartz.

**Data curation:** Michele M. Easter, Nicole L. Schramm-Sapyta.

**Formal analysis:** Michele M. Easter.

**Funding acquisition:** Nicole L. Schramm-Sapyta.

**Investigation:** Michele M. Easter.

**Methodology:** Michele M. Easter.

**Project administration:** Nicole L. Schramm-Sapyta.

**Resources:** Nicole L. Schramm-Sapyta.

**Supervision:** Maria A. Tackett.

**Writing – original draft:** Michele M. Easter.

**Writing – review & editing:** Nicole L. Schramm-Sapyta, Marvin S. Swartz, Maria A. Tackett, Lawrence H. Greenblatt.

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
