## [Decision Letter · Decision Letter 0]

31 Jul 2024

Primary care need and engagement by people with criminal legal involvement: Descriptive and associational analysis using retrospective data on the entire population ever detained in one southeastern U.S. county jail 2014-2020

PONE-D-24-22674

Dear Dr. Easter,

We’re pleased to inform you that your manuscript has been judged scientifically suitable for publication and will be formally accepted for publication once it meets all outstanding technical requirements.

Kind regards,

Souparno Mitra, M.D.

Academic Editor

PLOS ONE

Reviewers' comments:

Reviewer's Responses to Questions

**Comments to the Author**

1. Is the manuscript technically sound, and do the data support the conclusions?

Reviewer #1: Yes

Reviewer #2: Yes

2. Has the statistical analysis been performed appropriately and rigorously? 

Reviewer #1: Yes

Reviewer #2: Yes

3. Have the authors made all data underlying the findings in their manuscript fully available?

Reviewer #1: Yes

Reviewer #2: Yes

4. Is the manuscript presented in an intelligible fashion and written in standard English?

Reviewer #1: Yes

Reviewer #2: Yes

5. Review Comments to the Author

Reviewer #1: This article stresses the importance of promoting consistent engagement in primary care for individuals in the correctional system, especially those with chronic illnesses. The limitations of the study underscore the need for better disease measurement, comprehensive data collection, improved tracking of behavioral health services, support for continuity of care, broader study coverage, and use of data that is not tied to specific payers. Addressing these limitations will lead to a more precise understanding of the healthcare needs and usage trends of individuals in the correctional system, which will result in more targeted interventions and better health outcomes. Overall, it's an excellent article.

Reviewer #2: Thank you for the opportunity to review the manuscript titled "Primary care need and engagement by people with criminal legal involvement: Descriptive and associational analysis using retrospective data on the entire population ever detained in one southeastern U.S. County jail 2014-2020”. Well written manuscript, especially addressing healthcare needs in a particularly vulnerable population facing significant disparity in getting access to primary care.

6. PLOS authors have the option to publish the peer review history of their article (what does this mean?). If published, this will include your full peer review and any attached files.

Reviewer #1: **Yes: **Rajasekhar Kannali

Reviewer #2: No

---

## [Editor Report · Acceptance letter]

20 Aug 2024

PONE-D-24-22674 

PLOS ONE

Dear Dr. Easter, 

I'm pleased to inform you that your manuscript has been deemed suitable for publication in PLOS ONE. Congratulations! Your manuscript is now being handed over to our production team.

Kind regards, 

on behalf of

Dr. Souparno Mitra 

Academic Editor

PLOS ONE